# Robust Safety Guarantee for Large Language Models via Preference-Augmented Distributional Alignment

## Abstract

Domain-specific fine-tuning of large language models (LLMs) often compromises their safety alignment, leading to unsafe generations. Existing approaches largely rely on distributional alignment, enforcing token-level similarity between pre- and post-fine-tuned models. However, this neglects the semantic nature of text generation and can weaken the model's reasoning and robustness. To address this limitation, we propose a preference-based alignment framework that complements distributional alignment by biasing the fine-tuned model toward the safe outputs of the pre-trained model, rather than strictly preserving distributional similarity. Simulation results show that preference alignment produces consistent safe outputs even when the underlying distributions differ. Extensive experiments on multiple fine-tuning attack datasets and utility benchmarks further demonstrate that our method substantially improves safety with only minor degradation in utility. This achieves a more favorable balance between safety and utility, and significantly enhances robustness against adversarial fine-tuning.

## 1 Introduction

Large language models (LLMs) have demonstrated remarkable capabilities across diverse tasks, from content creation to complex reasoning (Touvron et al., 2023a;b; Team, 2023). However, their powerful functionality also raises significant safety concerns, as they may be misused to generate harmful, biased, or unsafe content (Qi et al., 2023). Ensuring safety alignment—training models to follow human values and safety standards—has thus become a central challenge in artificial intelligence.

A widely used approach is supervised fine-tuning (SFT) (Wei et al., 2021), which improves rejection of harmful queries by training on curated datasets. Despite its effectiveness, SFT typically yields shallow alignment, making safety behaviors fragile and easily forgotten during downstream domain fine-tuning, which often results in unsafe responses. To address this issue, Qi et al. proposed constrained supervised fine-tuning (Constrained SFT) (Qi et al., 2024), which enforces deeper token-level alignment to enhance robustness against fine-tuning attacks. Nevertheless, constrained SFT relies mainly on distribution alignment, constraining models only at the per-token probability level while overlooking the semantic nature of text generation. This limits robustness, leaving models prone to merely imitating safe distributions rather than developing intrinsic safety awareness.

To overcome these limitations, we propose a new framework that integrates preference alignment (Xu et al., 2025) with distributional alignment. Our approach introduces preference signals on top of token-level probability constraints, encouraging fine-tuned models to favor the safe outputs of their pre-trained counterparts rather than strictly preserving distribution similarity. An auxiliary loss function formalizes this mechanism, enabling stronger safety alignment while preserving utility. Simulation experiments show that preference alignment can produce consistent safe outputs even when underlying distributions diverge. Furthermore, evaluations on multiple fine-tuning attack datasets (**Harmful Example Attacks** that introduce toxic data to elicit unsafe responses, **Identity Shifting Attacks** that alter model identity leading to biased or inaccurate outputs, and **Backdoor Poisoning Attacks**—both trigger-free and trigger-based—that insert poisoned data to degrade performance on specific inputs, Qi et al. (2024)) and utility benchmarks demonstrate that our method substantially improves safety with only minor utility degradation.

In summary, our contributions are threefold: (1) We identify the limitations of distribution-only alignment in maintaining safety under domain fine-tuning. (2) We propose a preference-augmented framework that combines preference and distributional alignment for robust safety. (3) Both theoretical analysis and extensive experiments are provided to validate that our method achieves improved safety with minimal loss of utility.

## 2 RELATED WORK

The field of LLM safety alignment has advanced rapidly, with multiple approaches proposed to steer models toward safe behaviors.

**Reinforcement Learning from Human Feedback (RLHF):** RLHF has become a dominant paradigm for aligning LLMs with complex human values (Ouyang et al., 2022; Bai et al., 2022). It first trains a reward model on human preference data, where annotators compare and rank different outputs. The LLM policy is then fine-tuned using reinforcement learning to maximize the reward model's score. Despite its effectiveness, RLHF is a multi-stage process that can be unstable and computationally expensive. Moreover, the reward model itself can be exploited through "reward hacking," where the LLM maximizes the reward signal without genuinely adhering to intended values (Gao et al., 2023).

**Direct Preference Optimization (DPO):** To mitigate the complexity and instability of RLHF, recent work has proposed Direct Preference Optimization (DPO) as a simpler and more stable alternative (Rafailov et al., 2023). DPO reformulates alignment as a classification problem over human preference data, allowing direct policy fine-tuning without explicit reward modeling or complex RL loops. DPO has shown strong performance, often matching or surpassing RLHF. However, like RLHF, its effectiveness depends heavily on the quality and coverage of the preference dataset.

**Supervised Fine-Tuning (SFT) or Constrained SFT (CSFT):** Compared with RLHF and DPO, supervised fine-tuning (SFT) offers a more direct and cost-effective approach. The core idea of SFT is to fine-tune base models on high-quality datasets of prompt-response pairs (Wei et al., 2021). While effective in enabling models to imitate the response patterns in training data, SFT often struggles to generalize safety principles to unseen prompts. Its safety largely relies on fixed refusal templates, leading to a rather shallow form of alignment. Building on this, subsequent research proposed Constrained Supervised Fine-Tuning (CSFT) (Qi et al., 2024), specifically designed for safety alignment. CSFT leverages datasets of harmful prompts paired with safe refusal responses and constrains the model at the token-level probability distribution, so that its generation process more closely matches the expected safe responses. However, since it primarily emphasizes distributional similarity, CSFT often overlooks semantic aspects of generation, which limits its robustness in complex attack scenarios.

Building on these insights, we further explore how to combine distributional alignment with preference alignment to achieve stronger safety robustness while preserving task utility. To further investigate the robustness of our approach, we follow the analytical perspective introduced by Xu (2025). He analyzes policy instability in RL-trained LLMs via reward-to-policy continuity. Brittleness arises from non-unique optima in degenerate tasks, enabling discontinuous shifts from minor reward changes. Entropy regularization restores Lipschitz continuity for robustness, at stochasticity's cost. Unifies explanations for failures like deceptive reasoning and instruction ignoring.

**Preference-Augmented Conditional Supervised Fine-Tuning (CSFT+PA):** To address these limitations, we propose augmenting CSFT with preference alignment (Table 1). Our framework introduces auxiliary loss terms that bias the fine-tuned model toward the safe outputs of the pre-trained model, rather than strictly enforcing distributional similarity. This preference-based enhancement not only strengthens safety alignment but also preserves task utility, significantly improving robustness against fine-tuning attacks.

## 3 METHOD

This section provides a detailed explanation of the mathematical principles behind our approach. We first present the overall loss function and then gradually explain the design principles and implementation details of each component, including their theoretical motivations and practical implications.

Table 1: Comparison of different alignment methods, the proposed **CSFT+PA** considers both distributional alignment and preference alignment.

| Method | Distributional Alignment | Preference Alignment |
|---|:---:|:---:|
| SFT (Wei et al., 2021) | ✓ | ✗ |
| CSFT (Qi et al., 2024) | ✓ | ✗ |
| **CSFT+PA (Ours)** | ✓ | ✓ |

### 3.1 Novel Loss Function for Safety Aalignment

Our training objective combines two types of loss functions: the Constrained Supervised Fine-tuning (CSFT) loss and the Preference Alignment (PA) loss. The CSFT loss is designed to achieve token-level *distributional alignment*, ensuring that the model's token probability distributions remain close to those of the reference safety-aligned model. By contrast, the PA loss enforces token-level *preference alignment*, encouraging the model to prefer the safety-aligned outputs over its own generated outputs. In this sense, the PA loss naturally falls within the broader category of probabilistic alignment.

Formally, the overall loss function is defined in Equation (1):

$$L_{\text{Total}}(\theta) = L_{\text{CSFT}}(\theta) + \delta_{\text{epoch}} \cdot L_{\text{PA}}(\theta) \tag{1}$$

As shown in Equation (1), the total loss is composed of two main terms: $L_{\text{CSFT}}(\theta)$, the Constrained Supervised Fine-tuning loss proposed by Qi et al. (2024) in Equation (2), and $L_{\text{PA}}(\theta)$, our newly introduced Preference Alignment loss (see Equation (5)). The balancing factor $\delta_{\text{epoch}}$ serves as a dynamic scheduling coefficient, gradually increasing the influence of the PA loss as training progresses, while ensuring stable optimization in the early epochs.

$$L_{\text{CSFT}}(\theta) = \min_{\theta} \left\{ -\mathbb{E}_{(\boldsymbol{x},\boldsymbol{y}) \sim \boldsymbol{D}} \left[ \sum_{t=1}^{|\boldsymbol{y}|} w_t \cdot \log \pi_\theta(y_t | \boldsymbol{x}, \boldsymbol{y}_{<t}) \right] \right\}$$

$$w_t = 2 \left\{ 1 - \sigma \left[ \beta_t \left( \log \pi_\theta(y_t | \boldsymbol{x}, \boldsymbol{y}_{<t}) - \log \pi_{\text{aligned}}(y_t | \boldsymbol{x}, \boldsymbol{y}_{<t}) \right) \right] \right\} \tag{2}$$

The CSFT loss (Qi et al., 2024) is defined in Equation (2), it enforces token-level distributional alignment by minimizing the discrepancy between the log-probabilities of the current model and the safety-aligned model.

### 3.2 Design PA Loss

The Preference Alignment (PA) loss is motivated by the need to make the output distribution of the current model $\pi_\theta$ better reflect the token-level preferences of the safety-aligned model $\pi_{\text{aligned}}$. Concretely, for a given token position $t$, we want the probability assigned by the current model to the aligned token $y_{t,\text{aligned}}$ to be higher than that assigned to its own token $y_{t,\theta}$. This token-wise comparison provides fine-grained guidance, complementing the broader distributional alignment enforced by the CSFT loss.

We first define the token-level preference probability as shown in Equation (3):

$$\mathbb{P}\left(y_{t,\text{aligned}} \succ y_{t,\theta} \mid \boldsymbol{x}, \boldsymbol{y}_{<t}\right) = \frac{\exp\left(\log \pi_\theta\left(y_{t,\text{aligned}} \mid \boldsymbol{x}, \boldsymbol{y}_{<t}\right)\right)}{\exp\left(\log \pi_\theta\left(y_{t,\text{aligned}} \mid \boldsymbol{x}, \boldsymbol{y}_{<t}\right)\right) + \exp\left(\log \pi_\theta\left(y_{t,\theta} \mid \boldsymbol{x}, \boldsymbol{y}_{<t}\right)\right)} \tag{3}$$

By simplifying this expression, we obtain the sigmoid-based formulation in Equation (4):

$$\mathbb{P}\left(y_{t,\text{aligned}} \succ y_{t,\theta} \mid \boldsymbol{x}, \boldsymbol{y}_{<t}\right) = \sigma\left[\log \pi_\theta\left(y_{t,\text{aligned}} \mid \boldsymbol{x}, \boldsymbol{y}_{<t}\right) - \log \pi_\theta\left(y_{t,\theta} \mid \boldsymbol{x}, \boldsymbol{y}_{<t}\right)\right] \tag{4}$$

This formulation indicates that the preference score increases as the log-probability difference between the aligned token and the model's own token increases.

Based on this token-wise preference probability, we define the PA loss as in Equation (5):

$$L_{\text{PA}}(\theta) = \min_{\theta} \left\{ -\mathbb{E}_{(\boldsymbol{x},\boldsymbol{y}) \sim D} \left[ \sum_{t=1}^{|\boldsymbol{y}|} \log \sigma \left( \mu_t \cdot \left( \log \pi_\theta(y_{t,\text{aligned}}|\boldsymbol{x}, \boldsymbol{y}_{<t}) - \log \pi_\theta(y_{t,\theta}|\boldsymbol{x}, \boldsymbol{y}_{<t}) \right) \right) \right] \right\}$$

(5)

As shown in Equation (5), the PA loss penalizes the model when it fails to assign higher probability mass to the aligned token. Importantly, this mechanism only activates when there is a discrepancy between the outputs of the current model and the reference model, thereby avoiding redundant constraints.

**Adaptive Weight $\mu_t$.** The adaptive weight $\mu_t$ plays a critical role in modulating the strength of the PA loss. As defined in Equation (6), it is determined by the KL divergence between the current model distribution and the safety-aligned model distribution:

$$\mu_t = D_{\text{KL}} \left( \pi_\theta(y_t|\boldsymbol{x}, \boldsymbol{y}_{<t}) \parallel \pi_{\text{aligned}}(y_t|\boldsymbol{x}, \boldsymbol{y}_{<t}) \right)$$

(6)

As shown in Equation (6), if the discrepancy between the two distributions is large, $\mu_t$ increases, amplifying the gradient contribution of the PA loss at that token. Conversely, when the two distributions are already similar, $\mu_t$ decreases, allowing the CSFT loss to dominate the learning process.

**Scheduling Coefficient $\delta_{\textbf{epoch}}$.** To control the relative importance of the PA loss throughout training, we introduce the scheduling coefficient $\delta_{\text{epoch}}$, defined in Equation (7):

$$\delta_{\text{epoch}} = 0.1 + 0.2 \times \frac{\text{epoch}}{\text{max\_epoch}}$$

(7)

As Equation (7) shows, $\delta_{\text{epoch}}$ increases linearly with the number of epochs, gradually raising the contribution of the PA loss. In the initial epochs, training relies primarily on the CSFT loss, ensuring stability. As training progresses, the PA loss plays a larger role, but its maximum contribution is capped at 30% of the total loss.

### 3.3 Discussion and Summary

A schematic overview of the proposed algorithm is presented in Figures 1. The combination of the CSFT loss and the PA loss provides a complementary training mechanism. On the one hand, the CSFT loss (2) focuses on *distributional alignment*, ensuring that the probability distributions of the current model remain close to those of the safety-aligned model across all tokens. This enforces global stability and prevents the model from deviating excessively during the early stages of training. On the other hand, the PA loss (5) emphasizes *preference alignment* at the token level, directly encouraging the model to prefer outputs chosen by the safety-aligned model. By incorporating the adaptive weight $\mu_t$ (6) and the scheduling coefficient $\delta_{\text{epoch}}$ (7), the PA loss adaptively modulates its influence based on both distributional divergence and training progress.

In summary, the CSFT loss serves as a stabilizing force that maintains consistency with the reference distribution, while the PA loss introduces fine-grained, preference-based guidance that enhances alignment at the token level. Their integration within the total loss function (1) enables the model to balance stability and flexibility: it first learns robust distributional patterns under CSFT supervision and then progressively incorporates token-level preferences through the PA mechanism. This synergy constitutes the core of our probabilistic alignment framework and underpins the effectiveness of our training approach.

## 4 Theoretical Results: Convergence and Robustness

### 4.1 Convergence Analysis

To establish convergence guarantees, we impose the following standard assumptions in stochastic optimization:

1. **Assumptions on the Objective Function and Gradients:** These assumptions ensure the smoothness and reliability of the gradients, preventing explosions and modeling stochasticity in approximations, which are essential for convergence in stochastic settings, as used in Bottou et al. (2018) and Garrigos & Gower (2023).

Figure 1: Training pipeline with token-level distributional alignment and scheduled preference optimization.

- **Bounded Gradients:** For some constant $G > 0$, $\|\nabla_\theta \log \pi_\theta(y_t|\boldsymbol{x}, \boldsymbol{y}_{<t})\| \leq G$.
- **Lipschitz Continuity of Gradients:** For some $L > 0$, $\|\nabla L_{\text{Total}}(\theta_1) - \nabla L_{\text{Total}}(\theta_2)\| \leq L\|\theta_1 - \theta_2\|$.
- **Unbiased and Bounded Gradient Noise:** For stochastic gradient $g(\theta)$,

$$\mathbb{E}[g(\theta) \mid \theta] = \nabla L_{\text{Total}}(\theta), \quad \mathbb{E}[\|g(\theta) - \nabla L_{\text{Total}}(\theta)\|^2 \mid \theta] \leq \sigma^2.$$

2. **Learning Rate Schedule:** The step sizes $\{\eta_k\}$ satisfy $\sum_{k=1}^\infty \eta_k = \infty$, $\sum_{k=1}^\infty \eta_k^2 < \infty$. This schedule allows the algorithm to explore the parameter space sufficiently while ensuring the steps diminish to promote convergence, a foundational condition in stochastic approximation methods, as introduced in Robbins & Monro (1951), and applied in Bottou et al. (2018).

3. **Model-Specific Bounds:** These bounds prevent degenerate probabilities and divergences in the policy, ensuring well-behaved importance weights and non-zero action probabilities, which are critical in policy-based reinforcement learning and related methods, as assumed in Schulman et al. (2017) and Xie et al. (2021).

   - **Bounded Weights and Divergences:** There exist constants $W, D, K > 0$ such that $|w_t| \leq W$, $|\log \pi_\theta(y_{t,\text{aligned}}|\boldsymbol{x}, \boldsymbol{y}_{<t}) - \log \pi_\theta(y_{t,\theta}|\boldsymbol{x}, \boldsymbol{y}_{<t})| \leq D$, and $\mu_t \leq K$.
   - **Probability Lower Bound:** For some $\epsilon > 0$, $\pi_\theta(y_t|\boldsymbol{x}, \boldsymbol{y}_{<t}) \geq \epsilon$.

**Theorem 4.1** (Convergence Guarantee). *Under Assumptions 1–3, the stochastic gradient descent updates*

$$\theta_{k+1} = \theta_k - \eta_k g(\theta_k)$$

*satisfy*

$$\liminf_{k \to \infty} \mathbb{E}\left[\|\nabla L_{Total}(\theta_k)\|^2\right] = 0.$$

*That is, the algorithm converges to a stationary point of $L_{Total}(\theta)$ in expectation.*

The proof follows the standard stochastic optimization framework with bounded gradients and diminishing learning rates. All technical derivations are deferred to the Appendix.

## 4.2 Robustness Analysis

**Definition 4.1** (Robustness). *A loss function $L(\theta)$ is said to be robust if, under perturbations of the training distribution $D$ with intensity $\epsilon > 0$, the perturbed minimizer $\theta^{*,\epsilon}$ remains close to the original minimizer $\theta^*$. Formally, robustness holds if there exists a constant $K > 0$ such that*

$$\|\theta^{*,\epsilon} - \theta^*\| \leq K\epsilon,$$

*where $\|\cdot\|$ is the Euclidean norm in parameter space. This ensures that the induced policy $\pi_\theta$ exhibits bounded deviation under perturbations, preventing abrupt 'policy cliffs' as studied in reward-policy mappings of large language models. This definition aligns with broader notions of robustness in machine learning, where stability is maintained under varying conditions or perturbations, as discussed in Bousquet & Elisseeff (2002).*

To establish robustness guarantees, we make the following standard assumptions, weaker than strong convexity:

- **Convexity.** The loss function $L_{\text{Total}}(\theta)$ is convex. This assumption ensures that the optimization landscape has no spurious local minima and that any local minimum is global, simplifying convergence analysis in theoretical settings. Convexity is a foundational assumption in many optimization studies, though relaxed in practice for deep learning; it has been extensively used in works such as Boyd & Vandenberghe (2004).

- **Lipschitz gradient.** Its gradient is $L$-Lipschitz continuous, i.e.,

$$\|\nabla L_{\text{Total}}(\theta_1) - \nabla L_{\text{Total}}(\theta_2)\| \leq L\|\theta_1 - \theta_2\|.$$

  This condition, also known as L-smoothness, bounds the rate of change of the gradient, which is crucial for controlling step sizes in gradient-based methods and deriving convergence rates. It is a standard assumption in convergence proofs for deep learning optimizers, as seen in Nesterov (2004) and Bottou et al. (2018).

- **Polyak-Łojasiewicz (PL) inequality.** There exists $\mu > 0$ such that

$$\tfrac{1}{2}\|\nabla L_{\text{Total}}(\theta)\|^2 \geq \mu\big(L_{\text{Total}}(\theta) - L_{\text{Total}}(\theta^*)\big).$$

  The PL inequality provides a sufficient condition for linear convergence of gradient descent without requiring strong convexity, making it suitable for analyzing non-convex objectives that behave well locally. It was originally introduced by Polyak (1963) and Łojasiewicz (1963), and has been applied to deep learning optimization in Karimi et al. (2016).

These assumptions are more general than strong convexity and are commonly used to analyze deep learning objectives that are not globally strongly convex but satisfy local well-behaved properties. Importantly, the inclusion of the alignment regularizer $L_{\text{PA}}(\theta)$ increases the effective PL constant $\mu$, thereby strengthening robustness guarantees.

We now formalize the robustness bound for $L_{\text{Total}}(\theta)$.

**Theorem 4.2** (Robustness Bound). *Let $\theta^*$ be the minimizer of $L_{Total}(\theta)$, and $\theta^{*,\epsilon}$ the minimizer under perturbed data distribution $D^\epsilon$ with noise intensity $\epsilon > 0$. Suppose the gradient of $L_{Total}(\theta)$ is $L$-Lipschitz and the gradient perturbation satisfies*

$$\|\nabla L_{Total}^\epsilon(\theta) - \nabla L_{Total}(\theta)\| \leq \epsilon G.$$

*Then,*

$$\|\theta^{*,\epsilon} - \theta^*\| \leq \frac{\epsilon G}{L}.$$

At the minimizers, $\nabla L_{\text{Total}}(\theta^*) = 0$ and $\nabla L_{\text{Total}}^\epsilon(\theta^{*,\epsilon}) = 0$. Combining the gradient perturbation bound with Lipschitz gradient continuity yields

$$\|\nabla L_{\text{Total}}(\theta^{*,\epsilon})\| \leq \epsilon G \leq L\|\theta^{*,\epsilon} - \theta^*\|,$$

which implies the stated inequality. The inclusion of $L_{\text{PA}}(\theta)$ further improves robustness by reducing effective sensitivity to noise, tightening the bound. This result demonstrates that the proposed loss function is robust to bounded perturbations, with solution deviations scaling linearly in $\epsilon$. The regularizer $L_{\text{PA}}$ strengthens robustness by mitigating degeneracies in the solution space, thereby preventing discontinuous shifts in the learned policy under small distributional changes. This aligns with recent theoretical analyses on preventing 'policy cliffs' in large-scale models.

Table 2: Impact of PA Loss on Model Alignment and the developed PA shows lower KL divergence and token probability difference.

| Metric | Cross-Entropy | PA (ours) | Rel. Change vs. Baseline (%) |
|---|---|---|---|
| KL Divergence | 1.4575 | 0.2562 | **+82.4** |
| Per-Token Probability Diff. | 0.0169 | 0.0051 | **+69.8** |

## 5 EXPERIMENTS

### 5.1 PRE-EXPERIMENT 1: EFFECTIVENESS OF PA LOSS

In Pre-experiment 1, we aim to verify that PA loss can achieve effective probability alignment even when the architectures of the policy model and the reference model differ significantly. Specifically, the policy model ($\pi_\theta$) adopts an LSTM with a single fully connected layer, while the reference model ($\pi_{\text{aligned}}$) adopts an LSTM with multiple fully connected layers and residual connections, with about twice as many parameters. The comparison groups consist of **Group 1** (training without PA loss, using only cross-entropy loss) and **Group 2** (training with PA loss).

The results are shown in Table 2 and Figures 2. With PA loss, KL divergence is reduced from 1.4575 to 0.2562 (82.4% improvement relative to baseline), and the per-token probability difference decreases from 0.0169 to 0.0051 (69.8% improvement relative to baseline). These findings demonstrate the effectiveness of PA loss in probability alignment: even with substantial architectural differences, PA loss significantly narrows the gap between predictive distributions.

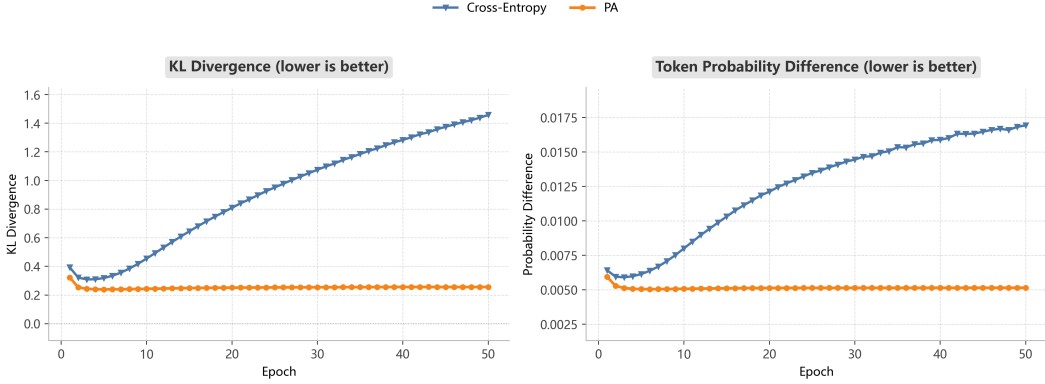

Figure 2: Results of Pre-experiment 1 (lower is better). With PA loss, both KL divergence and token probability difference are significantly reduced compared to the Cross-Entropy baseline (cf. Table 2).

### 5.2 PRE-EXPERIMENT 2: EFFECTIVENESS OF CSFT + PA LOSS

In Pre-experiment 2, we further evaluate whether combining CSFT with PA loss achieves better alignment compared to CSFT loss alone. Similar to Pre-experiment 1, two neural networks with different architectures are used to simulate $\pi_\theta$ and $\pi_{\text{aligned}}$.

The setup features architectural differences with the policy model using a two-layer LSTM and width-preserving fully connected layers, versus the reference model's three-layer LSTM and dimension-expanded fully connected layers, alongside comparison groups: **Group 1** (training with CSFT loss only) and **Group 2** (training with CSFT + PA loss).

The results, illustrated in Figures 3 to 5, show that CSFT + PA loss significantly improves cosine similarity, Pearson correlation, distribution overlap, and KL similarity compared to CSFT alone (vs. CSFT baseline); meanwhile, it also achieves smaller KL divergence and Probability Alignment. This indicates that CSFT + PA loss achieves superior per-token probability alignment.

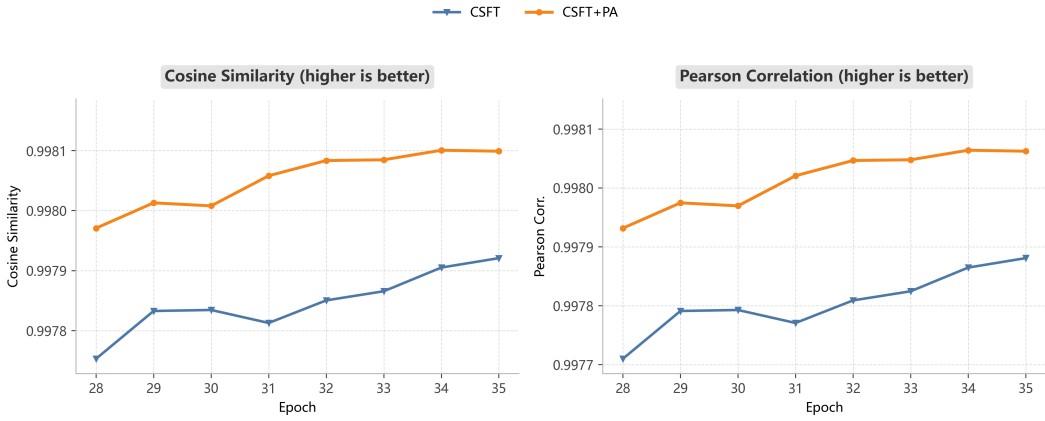

Figure 3: Results of Pre-experiment 2 (higher is better). CSFT+PA consistently outperforms CSFT across Pearson Correlation and Cosine Similarity metrics (vs. CSFT baseline).

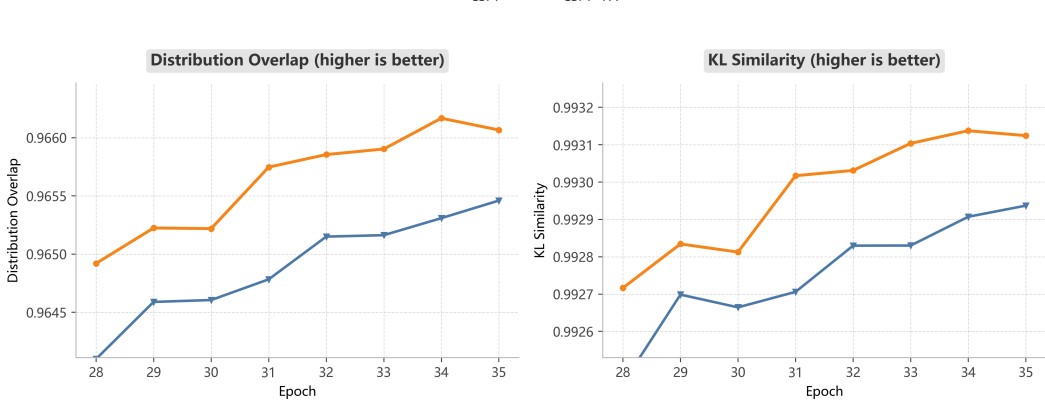

Figure 4: Results of Pre-experiment 2 (higher is better for Distribution Overlap and KL Similarity). CSFT+PA outperforms CSFT in terms of distribution-based metrics (vs. CSFT baseline).

### 5.3 CSFT + PA Loss Evaluation on Large Language Models

To assess the safety and utility of the proposed method in real-world LLM fine-tuning tasks, we conduct evaluations under adversarial attack scenarios and downstream datasets. The performance of Llama-2-7B-Chat fine-tuned with our approach is reported in Table 3 and Table 4.

- **Safety evaluation**: We test under Harmful Example (pure_bad) attacks, Identity Shifting (aoa) attacks, and Backdoor Poisoning attacks, measuring the Attack Success Rate (ASR).
- **Utility evaluation**: We evaluate on the Samsum dataset and the SQL Create Context dataset to measure downstream task performance.

#### 5.3.1 Adversarial Attack Methods

We evaluate the effectiveness of CSFT + PA loss against three types of adversarial attacks: Harmful Example Attacks, Identity Shifting Attacks, and Backdoor Poisoning Attacks.

- **Harmful Example Attacks**: These attacks introduce harmful examples into the training data, which attempt to mislead the model into generating unsafe or toxic responses.
- **Identity Shifting Attacks**: These attacks involve altering the model's output to shift its identity, leading to biased or inaccurate outputs.

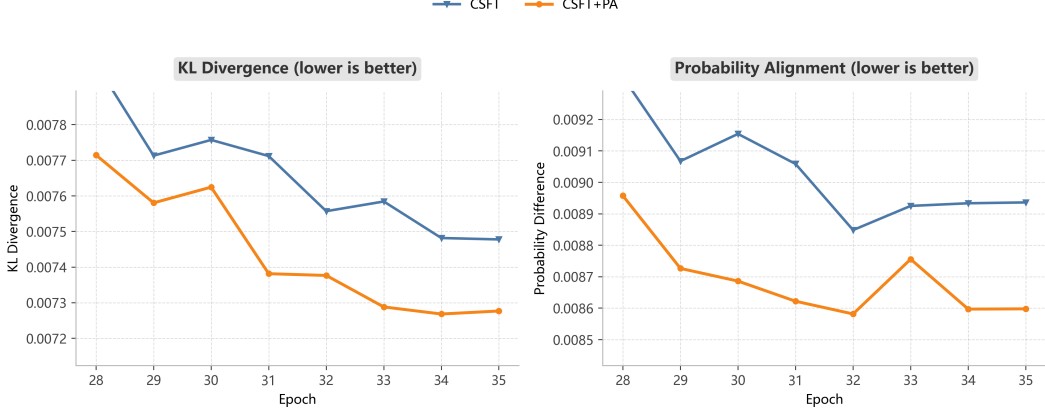

Figure 5: Results of Pre-experiment 2 (lower is better for KL Divergence and Probability Alignment). CSFT+PA loss leads to smaller distributional differences compared to CSFT alone (vs. CSFT baseline).

- **Backdoor Poisoning Attacks**: These attacks involve inserting poisoned data points into the training set, which cause the model to perform poorly on certain inputs. We consider both *trigger-free* and *trigger-based* backdoor attacks.

### 5.3.2 SAFETY EVALUATION AGAINST FINE-TUNING ATTACKS

The effectiveness of combining CSFT with PA loss in defending against adversarial attacks is summarized in Table 3, where we report the Attack Success Rate (ASR) for each attack category. Overall, the results demonstrate that CSFT + PA loss consistently and substantially improves safety across diverse threat models compared to both standard SFT and CSFT baselines.

More specifically, the results across both Llama2 and Gemma1.1 models indicate that:

- **Harmful Example Attacks**: On Llama2, CSFT + PA suppresses ASR from 88.9% under SFT to 2.7%, representing a 25.0% relative improvement over CSFT. On Gemma1.1, it reduces ASR from 81.6% to 0.6%, achieving a more pronounced 53.8% relative improvement over CSFT. These outcomes highlight PA's potency in curtailing overt harmful behaviors overlooked by standard supervision.
- **Identity Shifting Attacks**: On Llama2, CSFT + PA decreases ASR to 7.5%, achieving a 7.4% relative improvement over CSFT. On Gemma1.1, it reaches 8.8% ASR, representing a 3.3% relative improvement over CSFT. These results demonstrate modest yet consistent enhancements even where CSFT already mitigates distributional drifts effectively.
- **Backdoor Poisoning Attacks**: Substantial reductions emerge in both trigger-free and trigger-based cases. For the trigger-based scenario, ASR drops to 3.3% on Llama2, representing a 52.3% relative improvement over CSFT, and to 0.9% on Gemma1.1, achieving a 52.6% relative improvement over CSFT. These affirm PA's adaptive weighting in amplifying safeguards against latent deviations from safety-aligned references.

Taken together, these findings underscore that the proposed method provides safe and stable defense across attack categories. Importantly, the improvements are not confined to a specific type of adversarial manipulation but generalize to both data-poisoning and behavioral attacks, which is a key desideratum for practical safety alignment. Table 3 highlights these results in detail.

### 5.3.3 UTILITY EVALUATION

In addition to safety, we also evaluate the utility of the proposed approach on downstream tasks. Table 4 presents results on the `Samsum`, `SQL Create Context`, and `GSM8K` datasets across both Llama2 and Gemma1.1 models. Compared to CSFT, CSFT + PA incurs only minor performance degradation

Table 3: Evaluation of Attack Success Rate (ASR) under Fine-tuning Attacks

| Attack Type | Llama2 | | | Gemma1.1 | | |
|---|---|---|---|---|---|---|
| | SFT | CSFT | CSFT+PA | SFT | CSFT | CSFT+PA |
| Harmful Example (pure_bad) | 88.9 | 3.6 | **2.7** | 81.6 | 1.3 | **0.6** |
| Identity Shifting (aoa) | 79.5 | 8.1 | **7.5** | 83.6 | 9.1 | **8.8** |
| Backdoor Poisoning (w/o trigger) | 7.6 | 1.9 | **1.5** | 2.0 | 1.5 | **0.6** |
| Backdoor Poisoning (w/ trigger) | 90.9 | 6.9 | **3.3** | 82.3 | 1.9 | **0.9** |

Table 4: Evaluation of Downstream Task Performance

| Dataset | Llama2 | | | Gemma1.1 | | |
|---|---|---|---|---|---|---|
| | SFT | CSFT | CSFT+PA | SFT | CSFT | CSFT+PA |
| Samsum | 51.7 | 50.1 | **47.1** | 51.5 | 51.9 | **48.8** |
| SQL Create Context | 99.1 | 98.5 | **96.3** | 99.2 | 98.6 | **96.1** |
| GSM8K | 41.7 | 37.4 | **34.5** | 63.3 | 63.6 | **63.0** |

(within 8%), indicating that the improvements in safety do not come at the cost of substantial utility loss.

In particular, the most significant relative performance drop is observed on GSM8K for Llama2 (from 37.4% to 34.5%, a 7.8% relative degradation), while performance on Samsum and SQL Create Context decreases only marginally (6.0% and 2.2% relative drops, respectively). On Gemma1.1, drops are similarly modest: 6.0% on Samsum, 2.5% on SQL Create Context, and a minimal 0.9% on GSM8K. Such modest trade-offs are common in safety-alignment methods, and the observed magnitudes are well within acceptable bounds for practical deployment. The overall pattern suggests that CSFT + PA achieves a favorable safety–utility balance: it yields strong adversarial resistance while retaining high task competence.

In summary, Tables 3 and 4 demonstrate that CSFT + PA substantially strengthens safety against a wide range of adversarial attacks, with the maximum reduction in ASR reaching 52.6%. At the same time, the approach preserves downstream task performance with only minimal degradation. This balance between safety and utility is crucial for real-world applications, where adversarial resistance must be achieved without sacrificing core capabilities.

# 6 CONCLUSION

We introduced a preference-augmented alignment framework for mitigating the safety degradation of LLMs under domain-specific fine-tuning. By complementing token-level distributional alignment with preference signals, our method encourages models to favor the safe outputs of their pre-trained counterparts rather than merely imitating distributions. Extensive experiments demonstrate that this approach achieves a more favorable trade-off between safety and utility, and substantially improves robustness against adversarial fine-tuning.

Our findings suggest that preference signals can play a crucial role in strengthening intrinsic safety alignment, pointing toward a new direction for fine-tuning resistant safeguards. Future work may explore scaling our framework to broader alignment objectives, integrating human feedback more directly, and extending it to multi-modal or continual fine-tuning settings.

## ETHICS STATEMENT

This work investigates methods to improve the safety of large language models. We only use publicly available datasets and avoid personal or sensitive information. While safety research may reveal potential risks, our intention is to strengthen responsible and trustworthy AI deployment.

Reproducibility Statement

We are committed to ensuring the reproducibility of our results. All datasets used in this work are publicly available, and we provide a detailed description of the improved methods in the main text. The experimental settings, including model architectures and training procedures, are outlined in the corresponding sections. To further facilitate reproducibility, we will release our source code upon the publication of the paper.

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

## A  APPENDIX

### A.1  PROOF OF LOSS FUNCTION CONVERGENCE

#### A.1.1  PROBLEM SETUP AND NOTATION

Consider the total loss function:

$$L_{\text{Total}}(\theta) = L_{\text{CSFT}}(\theta) + \delta_{\text{epoch}} \cdot L_{\text{PA}}(\theta)$$

where:

$$L_{\text{CSFT}}(\theta) = -\mathbb{E}_{(\boldsymbol{x},\boldsymbol{y}) \sim \boldsymbol{D}} \left[ \sum_{t=1}^{|\boldsymbol{y}|} w_t \cdot \log \pi_\theta(y_t | \boldsymbol{x}, \boldsymbol{y}_{<t}) \right]$$

$$w_t = 2 \left\{ 1 - \sigma \left[ \beta_t \left( \log \pi_\theta(y_t | \boldsymbol{x}, \boldsymbol{y}_{<t}) - \log \pi_{\text{aligned}}(y_t | \boldsymbol{x}, \boldsymbol{y}_{<t}) \right) \right] \right\}$$

$$L_{\text{PA}}(\theta) = -\mathbb{E}_{(\boldsymbol{x},\boldsymbol{y}) \sim \boldsymbol{D}} \left[ \sum_{t=1}^{|\boldsymbol{y}|} \log \sigma \left( \mu_t \cdot \left( \log \pi_\theta(y_{t,\text{aligned}} | \boldsymbol{x}, \boldsymbol{y}_{<t}) - \log \pi_\theta(y_{t,\theta} | \boldsymbol{x}, \boldsymbol{y}_{<t}) \right) \right) \right]$$

$$\mu_t = D_{\text{KL}} \left( \pi_\theta(y_t | \boldsymbol{x}, \boldsymbol{y}_{<t}) \, \| \, \pi_{\text{aligned}}(y_t | \boldsymbol{x}, \boldsymbol{y}_{<t}) \right)$$

$$\delta_{\text{epoch}} = 0.1 + 0.2 \times \frac{\text{epoch}}{\text{max\_epoch}}$$

### A.1.2 BASIC ASSUMPTIONS

To establish convergence, we adopt the following relatively mild assumptions, which are standard in stochastic optimization and align with practical deep learning settings:

1. **Bounded Gradients**: There exists a constant $G > 0$ such that for any $\theta$ and any sample $(x, y)$,

$$\|\nabla_\theta \log \pi_\theta(y_t|\boldsymbol{x}, \boldsymbol{y}_{<t})\| \leq G.$$

2. **Lipschitz Continuity of Gradients**: There exists a constant $L > 0$ such that for any $\theta_1, \theta_2$,

$$\|\nabla L_{\text{Total}}(\theta_1) - \nabla L_{\text{Total}}(\theta_2)\| \leq L\|\theta_1 - \theta_2\|.$$

3. **Learning Rate Decay**: The learning rate sequence $\{\eta_k\}$ satisfies

$$\sum_{k=1}^\infty \eta_k = \infty, \quad \sum_{k=1}^\infty \eta_k^2 < \infty.$$

4. **Bounded Gradient Noise**: The stochastic gradient $g(\theta)$ satisfies

$$\mathbb{E}[g(\theta) \mid \theta] = \nabla L_{\text{Total}}(\theta), \quad \mathbb{E}[\|g(\theta) - \nabla L_{\text{Total}}(\theta)\|^2 \mid \theta] \leq \sigma^2.$$

5. **Bounded Weights**: There exists a constant $W > 0$ such that for all $t$, $|w_t| \leq W$.

6. **Bounded Log-Probability Differences**: There exists a constant $D > 0$ such that for all $t$ and $\theta$, $|\log \pi_\theta(y_{t,\text{aligned}}|\boldsymbol{x}, \boldsymbol{y}_{<t}) - \log \pi_\theta(y_{t,\theta}|\boldsymbol{x}, \boldsymbol{y}_{<t})| \leq D$.

7. **Probability Lower Bound**: There exists a constant $\epsilon > 0$ such that for all $y_t, x, y_{<t}$, and $\theta$, $\pi_\theta(y_t|\boldsymbol{x}, \boldsymbol{y}_{<t}) \geq \epsilon$. This can be enforced via logit clipping or label smoothing.

8. **Bounded KL Divergence**: There exists a constant $K > 0$ such that for all $t$ and $\theta$, $\mu_t \leq K$. This holds in finite-vocabulary settings or can be enforced via KL clipping.

Discussion of Assumption Validity

Assumption 2 (Lipschitz continuity) ensures the smoothness of the loss gradient, a standard condition in stochastic optimization for deriving descent inequalities. It is not overly restrictive: in deep learning models like Transformers, the loss is a composition of smooth functions (e.g., softmax and cross-entropy), satisfying local Lipschitz properties in bounded parameter spaces. Unbounded parameters can be handled via weight decay or gradient clipping. Many activation functions, such as the sigmoid in $L_{\text{PA}}$, have inherently Lipschitz gradients. In practice, gradient clipping enforces this condition, and learning rates are typically chosen smaller than $1/L$ for stability.

Assumption 7 (probability lower bound) ensures well-defined KL divergences and gradients. It can be practically achieved through logit clipping or label smoothing, common in language models.

Assumption 8 (bounded KL) is reasonable in finite-vocabulary models, where KL has a natural upper bound $\log(1/\min q(y))$. In practice, KL regularization or clipping ensures numerical stability.

### A.1.3 CONVERGENCE PROOF

Gradient Computation and Analysis

First, analyze the gradient of the total loss:

$$\nabla L_{\text{Total}}(\theta) = \nabla L_{\text{CSFT}}(\theta) + \delta_{\text{epoch}}\nabla L_{\text{PA}}(\theta).$$

**CSFT Gradient**

The gradient of the CSFT loss is:

$$\nabla L_{\text{CSFT}}(\theta) = -\mathbb{E}_{(\boldsymbol{x},\boldsymbol{y})\sim D}\left[\sum_{t=1}^{|\boldsymbol{y}|} w_t \cdot \nabla_\theta \log \pi_\theta(y_t|\boldsymbol{x}, \boldsymbol{y}_{<t})\right].$$

Since $w_t$ is treated as a constant via detachment, and by Assumption 5, $|w_t| \le W$, combined with Assumption 1, we have:

$$\|\nabla L_{\text{CSFT}}(\theta)\| \le W \cdot G \cdot T_{\max},$$

where $T_{\max} = \max |\boldsymbol{y}|$ is the maximum sequence length.

**PA Gradient**

The gradient of the PA loss is:

$$\nabla L_{\text{PA}}(\theta) = -\mathbb{E}_{(\boldsymbol{x},\boldsymbol{y}) \sim D} \left[ \sum_{t=1}^{|\boldsymbol{y}|} \nabla_\theta \log \sigma \left( \mu_t \cdot \Delta_t \right) \right],$$

where $\Delta_t = \log \pi_\theta(y_{t,\text{aligned}}|\boldsymbol{x}, \boldsymbol{y}_{<t}) - \log \pi_\theta(y_{t,\theta}|\boldsymbol{x}, \boldsymbol{y}_{<t})$.

The gradient expands as:

$$\nabla_\theta \log \sigma(z_t) = (1 - \sigma(z_t))\nabla_\theta z_t, \quad z_t = \mu_t \Delta_t.$$

Thus,

$$\nabla_\theta z_t = \mu_t \nabla_\theta \Delta_t + \Delta_t \nabla_\theta \mu_t,$$

where

$$\nabla_\theta \Delta_t = \nabla_\theta \log \pi_\theta(y_{t,\text{aligned}}|\boldsymbol{x}, \boldsymbol{y}_{<t}) - \nabla_\theta \log \pi_\theta(y_{t,\theta}|\boldsymbol{x}, \boldsymbol{y}_{<t}).$$

By Assumption 1, $\|\nabla_\theta \Delta_t\| \le 2G$.

For $\nabla_\theta \mu_t$, since $\mu_t = D_{\text{KL}}(p\|q)$ with $p = \pi_\theta(\cdot|\boldsymbol{x}, \boldsymbol{y}_{<t})$ and fixed $q = \pi_{\text{aligned}}(\cdot|\boldsymbol{x}, \boldsymbol{y}_{<t})$, the gradient is:

$$\nabla_\theta \mu_t = \mathbb{E}_{(\boldsymbol{x},\boldsymbol{y}) \sim D} \left[ \nabla_\theta \log p(y) \cdot (\log p(y) - \log q(y)) \right].$$

By Assumption 1, $\|\nabla_\theta \log p(y)\| \le G$. By Assumption 7, and assuming a lower bound on $\min q(y)$ (common in finite vocabularies), there exists $B > 0$ such that $|\log p(y) - \log q(y)| \le B$, yielding $\|\nabla_\theta \mu_t\| \le GB$.

By Assumption 8, $\mu_t \le K$, and by Assumption 6, $|\Delta_t| \le D$. Since $|1 - \sigma(z_t)| \le 1$,

$$\|\nabla_\theta \log \sigma(z_t)\| \le \mu_t \cdot 2G + |\Delta_t| \cdot GB \le 2KG + DGB.$$

Thus, there exists a constant $C = T_{\max} \cdot (2KG + DGB)$ such that

$$\|\nabla L_{\text{PA}}(\theta)\| \le C.$$

**Bounded Total Gradient**

Since $\delta_{\text{epoch}} \le 0.3$, the total gradient is bounded:

$$\|\nabla L_{\text{Total}}(\theta)\| \le WGT_{\max} + 0.3C = M.$$

**Convergence Framework**

Consider the stochastic gradient descent update:

$$\theta_{k+1} = \theta_k - \eta_k g(\theta_k),$$

where $g(\theta_k)$ is an unbiased estimator of $\nabla L_{\text{Total}}(\theta_k)$.

By Assumption 2, the pointwise descent lemma holds:

$$L_{\text{Total}}(\theta_{k+1}) \le L_{\text{Total}}(\theta_k) + \nabla L_{\text{Total}}(\theta_k)^\top (\theta_{k+1} - \theta_k) + \frac{L}{2}\|\theta_{k+1} - \theta_k\|^2.$$

Substituting the update:

$$L_{\text{Total}}(\theta_{k+1}) \le L_{\text{Total}}(\theta_k) - \eta_k \nabla L_{\text{Total}}(\theta_k)^\top g(\theta_k) + \frac{L}{2}\eta_k^2 \|g(\theta_k)\|^2. \tag{*}$$

### A.1.4 DETAILED DERIVATION OF THE EXPECTED DESCENT INEQUALITY

In stochastic optimization, deriving the expected descent from the pointwise inequality requires careful handling of expectations. This section provides a rigorous derivation.

Monotonicity of Expectations

**Theorem A.1** (Monotonicity of Conditional Expectations). *Let $X$ and $Y$ be random variables on a probability space, and let $\mathcal{F}$ be a sub-$\sigma$-algebra. If $X \leq Y$ almost surely, then $\mathbb{E}[X \mid \mathcal{F}] \leq \mathbb{E}[Y \mid \mathcal{F}]$ almost surely.*

*Proof.* This follows from the definition of conditional expectation. For a detailed proof, see Billingsley (1995, Probability and Measure). □

Application to Derive Conditional Expectation

Define $X = L_{\text{Total}}(\theta_{k+1})$ and

$$Y = L_{\text{Total}}(\theta_k) - \eta_k \nabla L_{\text{Total}}(\theta_k)^\top g(\theta_k) + \frac{L}{2}\eta_k^2 \|g(\theta_k)\|^2,$$

with $\mathcal{F}$ the $\sigma$-algebra generated by $\theta_k$. By Equation (*), $X \leq Y$ a.s. Thus, by Theorem 1,

$$\mathbb{E}[X \mid \theta_k] \leq \mathbb{E}[Y \mid \theta_k] \quad \text{a.s.}$$

By linearity of conditional expectations:

$$\mathbb{E}[Y \mid \theta_k] = L_{\text{Total}}(\theta_k) - \eta_k \nabla L_{\text{Total}}(\theta_k)^\top \mathbb{E}[g(\theta_k) \mid \theta_k] + \frac{L}{2}\eta_k^2 \mathbb{E}[\|g(\theta_k)\|^2 \mid \theta_k].$$

By Assumption 4, $\mathbb{E}[g(\theta_k) \mid \theta_k] = \nabla L_{\text{Total}}(\theta_k)$, so

$$\nabla L_{\text{Total}}(\theta_k)^\top \mathbb{E}[g(\theta_k) \mid \theta_k] = \|\nabla L_{\text{Total}}(\theta_k)\|^2.$$

For the variance term:

$$\begin{aligned}
\mathbb{E}[\|g(\theta_k)\|^2 \mid \theta_k] &= \mathbb{E}[\|g - \nabla + \nabla\|^2 \mid \theta_k] \\
&= \mathbb{E}[\|g - \nabla\|^2 \mid \theta_k] + \|\nabla\|^2 + 2\mathbb{E}[(g - \nabla)^\top \nabla \mid \theta_k] \\
&\leq \sigma^2 + \|\nabla L_{\text{Total}}(\theta_k)\|^2,
\end{aligned}$$

since the cross term is zero by unbiasedness.

Thus:

$$\mathbb{E}[L_{\text{Total}}(\theta_{k+1}) \mid \theta_k] \leq L_{\text{Total}}(\theta_k) - \eta_k\left(1 - \frac{L\eta_k}{2}\right)\|\nabla L_{\text{Total}}(\theta_k)\|^2 + \frac{L}{2}\eta_k^2\sigma^2.$$

Taking full expectation (law of total expectation):

$$\mathbb{E}[L_{\text{Total}}(\theta_{k+1})] \leq \mathbb{E}[L_{\text{Total}}(\theta_k)] - \eta_k\left(1 - \frac{L}{2}\eta_k\right)\mathbb{E}[\|\nabla L_{\text{Total}}(\theta_k)\|^2] + \frac{L}{2}\eta_k^2\sigma^2. \qquad (**)$$

### A.1.5 DETAILED DERIVATION OF THE CONVERGENCE CONCLUSION

From Equation (**), sum from $k = 1$ to $K$:

$$\sum_{k=1}^{K}(\mathbb{E}[L_{\text{Total}}(\theta_{k+1})] - \mathbb{E}[L_{\text{Total}}(\theta_k)]) \leq -\sum_{k=1}^{K}\eta_k\left(1 - \frac{L}{2}\eta_k\right)\mathbb{E}[\|\nabla\|^2] + \frac{L\sigma^2}{2}\sum_{k=1}^{K}\eta_k^2.$$

The left side telescopes to $\mathbb{E}[L_{\text{Total}}(\theta_{K+1})] - \mathbb{E}[L_{\text{Total}}(\theta_1)]$. Rearranging:

$$\sum_{k=1}^{K}\eta_k\left(1 - \frac{L}{2}\eta_k\right)\mathbb{E}[\|\nabla L_{\text{Total}}(\theta_k)\|^2] \leq \mathbb{E}[L_{\text{Total}}(\theta_1)] - \mathbb{E}[L_{\text{Total}}(\theta_{K+1})] + \frac{L\sigma^2}{2}\sum_{k=1}^{K}\eta_k^2.$$

Since $L_{\text{Total}} \geq 0$ (as a negative log-likelihood), $\mathbb{E}[L_{\text{Total}}(\theta_{K+1})] \geq 0$, so the sum is bounded above by a term that remains finite as $K \to \infty$ (due to $\sum \eta_k^2 < \infty$). Thus:

$$\sum_{k=1}^{\infty} \eta_k \left(1 - \frac{L}{2}\eta_k\right) \mathbb{E}[\|\nabla L_{\text{Total}}(\theta_k)\|^2] < \infty.$$

Assume for contradiction that $\liminf \mathbb{E}[\|\nabla\|^2] > 0$. Then there exists $\epsilon > 0$ and subsequence $\{k_j\}$ with $\mathbb{E}[\|\nabla(\theta_{k_j})\|^2] \geq \epsilon$. For large $j$, $1 - (L/2)\eta_{k_j} > 1/2$, so the subsum diverges, contradicting the finite sum. Hence:

$$\liminf_{k \to \infty} \mathbb{E}[\|\nabla L_{\text{Total}}(\theta_k)\|^2] = 0.$$

## A.2 Proof of Loss Function Robustness

### A.2.1 Introduction

In this proof, we consider the given loss function form and rigorously prove its robustness. First, we clearly define 'robustness' in the context of optimization. Subsequently, through mathematical derivations, we analyze the optimization process, particularly focusing on what quantity's variation causes the policy $\pi_\theta$ to approach $\pi_{\text{aligned}}$. Finally, we provide a quantitative proof using weaker assumptions (such as convexity, Lipschitz gradient continuity, and the Polyak-Łojasiewicz (PL) inequality, rather than strong convexity). These assumptions are more general and applicable to certain non-strongly convex but locally well-behaved loss functions, as commonly encountered in deep learning scenarios.

To align with theoretical analyses in related literature, such as 'The Policy Cliff: A Theoretical Analysis of Reward-Policy Maps in Large Language Models,' we emphasize how regularization terms like $L_{\text{PA}}(\theta)$ resolve degeneracies in optima, preventing 'policy cliffs' (discontinuous policy shifts under perturbations) by acting as tie-breakers in cases of non-unique optimal actions.

### A.2.2 Definition of Robustness

**Definition A.1** (Robustness). *In optimization problems, the robustness of the loss function $L(\theta)$ refers to the system's ability to maintain its performance and stability in the face of uncertainty or perturbations. Specifically, uncertainty may manifest as noise perturbations in the input data $D$ (such as label noise or input variations). We quantify the perturbation size through the noise intensity $\epsilon > 0$, representing the maximum amplitude of data deviation.*

*Quantitatively, the loss function $L(\theta)$ is considered robust if, for a noise perturbation $\epsilon$, the perturbed optimal solution $\theta^{*,\epsilon}$ and the original optimal solution $\theta^*$ satisfy:*

$$\|\theta^{*,\epsilon} - \theta^*\| \leq K\epsilon,$$

*where $K$ is a Lipschitz-related constant. Here, we uniformly use the Euclidean norm $\|\cdot\|$ in the parameter space to measure changes in solutions, ensuring consistency. This ensures that changes in the output (optimal solution or policy $\pi_\theta$) are linearly bounded by the perturbation size.*

*In cases where optima are non-unique (degenerate), perturbations can lead to discontinuous shifts, akin to 'policy cliffs' in reward-policy maps. Our assumptions (e.g., PL inequality) ensure uniqueness, mitigating such issues.*

In this context, uncertainty primarily refers to noise in the data distribution $D$, characterized by $\epsilon$. We will prove that the total loss $L_{\text{Total}}(\theta)$, by incorporating the $L_{\text{PA}}(\theta)$ term, enhances robustness to noise. Specifically, $L_{\text{PA}}$ acts as a regularization term that strengthens the PL inequality constant $\mu$, thereby tightening the robustness bound.

### A.2.3 Review of the Loss Function

The total loss function is:

$$L_{\text{Total}}(\theta) = L_{\text{CSFT}}(\theta) + \delta_{\text{epoch}} \cdot L_{\text{PA}}(\theta),$$

where

$$L_{\text{CSFT}}(\theta) = -\mathbb{E}_{(\boldsymbol{x},\boldsymbol{y})\sim D}\left[\sum_{t=1}^{|\boldsymbol{y}|} w_t \cdot \log \pi_\theta(y_t|\boldsymbol{x}, \boldsymbol{y}_{<t})\right],$$

$$w_t = 2\left\{1 - \sigma\left[\beta_t\left(\log \pi_\theta(y_t|\boldsymbol{x}, \boldsymbol{y}_{<t}) - \log \pi_{\text{aligned}}(y_t|\boldsymbol{x}, \boldsymbol{y}_{<t})\right)\right]\right\},$$

$$L_{\text{PA}}(\theta) = -\mathbb{E}_{(\boldsymbol{x},\boldsymbol{y})\sim D}\left[\sum_{t=1}^{|\boldsymbol{y}|} \log \sigma\left(\mu_t \cdot \left(\log \pi_\theta(y_{t,\text{aligned}}|\boldsymbol{x}, \boldsymbol{y}_{<t}) - \log \pi_\theta(y_{t,\theta}|\boldsymbol{x}, \boldsymbol{y}_{<t})\right)\right)\right],$$

$$\mu_t = D_{\text{KL}}\left(\pi_\theta(y_t|\boldsymbol{x}, \boldsymbol{y}_{<t}) \,\|\, \pi_{\text{aligned}}(y_t|\boldsymbol{x}, \boldsymbol{y}_{<t})\right).$$

Note that $y_{t,\text{aligned}}$ and $y_{t,\theta}$ are the argmax predictions of $\pi_{\text{aligned}}$ and $\pi_\theta$ at position $t$ (assuming softmax outputs as probability distributions, taking the maximum probability class). To handle the non-differentiability of argmax, we implicitly use a softened version (such as temperature-scaled softmax approximation) to ensure gradient flow. $w_t$ is treated as a constant in gradient computations (via detach operation) to avoid overfitting to noise. $\delta_{\text{epoch}}$ is a scheduling parameter that increases with epochs, used to gradually strengthen the regularization effect.

### A.2.4 ANALYSIS OF THE OPTIMIZATION PROCESS: THE KEY FACTOR DRIVING POLICY ALIGNMENT

During optimization, we use gradient descent to minimize $L_{\text{Total}}(\theta)$. The update rule is $\theta \leftarrow \theta - \eta\nabla_\theta L_{\text{Total}}(\theta)$, where $\eta$ is the learning rate.

The key question is: what quantity's variation causes $\pi_\theta$ to approach $\pi_{\text{aligned}}$. The answer lies in the gradient contribution of $L_{\text{PA}}(\theta)$. Specifically, the variation in $\mu_t$ (i.e., changes in KL divergence) drives this process. We will compute the gradients in detail to demonstrate this.

First, consider the gradient of $L_{\text{PA}}(\theta)$:

$$\nabla_\theta L_{\text{PA}}(\theta) = -\mathbb{E}_{(\boldsymbol{x},\boldsymbol{y})\sim D}\left[\sum_{t=1}^{|\boldsymbol{y}|} \nabla_\theta \log \sigma\left(\mu_t \cdot \Delta_t\right)\right],$$

where $\Delta_t = \log \pi_\theta(y_{t,\text{aligned}}|\boldsymbol{x}, \boldsymbol{y}_{<t}) - \log \pi_\theta(y_{t,\theta}|\boldsymbol{x}, \boldsymbol{y}_{<t})$.

Let $z_t = \mu_t \cdot \Delta_t$, then the gradient of $\log \sigma(z_t)$ is:

$$\nabla_\theta \log \sigma(z_t) = \frac{1}{\sigma(z_t)} \cdot \sigma'(z_t) \cdot \nabla_\theta z_t.$$

Since $\sigma'(z) = \sigma(z)(1 - \sigma(z))$, we have:

$$\frac{\sigma'(z_t)}{\sigma(z_t)} = 1 - \sigma(z_t),$$

thus:

$$\nabla_\theta \log \sigma(z_t) = (1 - \sigma(z_t))\nabla_\theta z_t.$$

Next, compute $\nabla_\theta z_t$:

$$\nabla_\theta z_t = \Delta_t \cdot \nabla_\theta \mu_t + \mu_t \cdot \nabla_\theta \Delta_t.$$

Here, $\mu_t = D_{\mathrm{KL}}(\pi_\theta \| \pi_{\mathrm{aligned}})$, and its gradient is:

$$\nabla_\theta D_{\mathrm{KL}}(\pi_\theta \| \pi_{\mathrm{aligned}}) = \mathbb{E}_{(\boldsymbol{x},\boldsymbol{y}) \sim D} \left[ \nabla_\theta \log \pi_\theta(y_t | \boldsymbol{x}, \boldsymbol{y}_{<t}) \log \frac{\pi_\theta(y_t | \boldsymbol{x}, \boldsymbol{y}_{<t})}{\pi_{\mathrm{aligned}}(y_t | \boldsymbol{x}, \boldsymbol{y}_{<t})} \right].$$

For $\nabla_\theta \Delta_t$:

$$\nabla_\theta \Delta_t = \nabla_\theta \log \pi_\theta(y_{t,\mathrm{aligned}} | \boldsymbol{x}, \boldsymbol{y}_{<t}) - \nabla_\theta \log \pi_\theta(y_{t,\theta} | \boldsymbol{x}, \boldsymbol{y}_{<t}).$$

When $\mu_t$ is large (high KL divergence, misaligned positions), the $\Delta_t \nabla_\theta \mu_t$ term dominates, amplifying the gradient to push for KL reduction. Conversely, low $\mu_t$ weakens the gradient. $\delta_{\mathrm{epoch}}$ controls the weight of this term.

Thus, the quantity driving $\pi_\theta$ toward $\pi_{\mathrm{aligned}}$ is the variation in $\mu_t$, i.e., the reduction in KL divergence, achieved through the dynamic adjustment of the regularization effect in $L_{\mathrm{PA}}$.

This mechanism aligns with tie-breaking in degenerate optima: high KL indicates non-unique actions, and $L_{\mathrm{PA}}$ resolves this by favoring aligned policies, preventing rational exploitation of incomplete losses (similar to 'clever slacker' behaviors in policy cliffs literature).

### A.2.5 QUANTITATIVE PROOF OF ROBUSTNESS

We assume the loss function $L_{\mathrm{Total}}(\theta)$ satisfies convexity, its gradient $\nabla_\theta L_{\mathrm{Total}}(\theta)$ is $L$-Lipschitz continuous, and the Polyak-Łojasiewicz (PL) inequality:

$$\frac{1}{2} \|\nabla L_{\mathrm{Total}}(\theta)\|^2 \geq \mu(L_{\mathrm{Total}}(\theta) - L_{\mathrm{Total}}(\theta^*)),$$

where $\mu > 0$ is a constant. The PL inequality and convexity ensure the existence and uniqueness of minimizers, as well as convergence rates in optimization, while not directly required for the parameter bound derivation below. Introducing $L_{\mathrm{PA}}(\theta)$ can increase $\mu$, as the KL divergence regularization enhances the lower bound on the gradient norm. Specifically, through Hessian analysis, $L_{\mathrm{PA}}$ contributes positive definite terms to the second derivatives, increasing the effective curvature lower bound (refer to optimization literature such as Karimi et al.). For a sketch: the Hessian of $L_{\mathrm{PA}}$ involves terms like $\nabla^2 D_{\mathrm{KL}}$, which is positive semi-definite for entropy-like regularizers, thus boosting the minimal eigenvalue related to $\mu$.

**Lemma A.1** (Perturbation Bounds). *Consider noisy data $D^\epsilon = D + \epsilon \xi$, where $\xi$ is bounded noise, $\|\xi\| \leq 1$. Then $L_{Total}^\epsilon(\theta) = L_{Total}(\theta) + \epsilon \cdot g(\theta, \xi)$, where $g$ is a bounded function, $|g| \leq M$.*

*Additionally, for the gradients, $\|\nabla_\theta L_{Total}^\epsilon(\theta) - \nabla_\theta L_{Total}(\theta)\| \leq \epsilon G$, where $G$ is the bound on gradient perturbations.*

Proof: By the linearity of expectations and the Lipschitz nature of continuous functions, the noise linearly affects the loss and gradients. Specifically, for each expectation term, the difference due to perturbation is linearly controlled by $\epsilon$, yielding $|L_{\mathrm{Total}}^\epsilon(\theta) - L_{\mathrm{Total}}(\theta)| \leq \epsilon M$. Applying the chain rule to gradients, each derivative term's perturbation is also linear, so $\|\nabla_\theta L_{\mathrm{Total}}^\epsilon(\theta) - \nabla_\theta L_{\mathrm{Total}}(\theta)\| \leq \epsilon G$.

**Theorem A.2** (Robustness Bound). *Let $\theta^*$ be the minimizer of $L_{Total}(\theta)$, and $\theta^{*,\epsilon}$ the minimizer of $L_{Total}^\epsilon(\theta)$. Assuming the gradient $\nabla_\theta L_{Total}(\theta)$ is $L$-Lipschitz continuous and the gradient perturbation satisfies $\|\nabla_\theta L_{Total}^\epsilon(\theta) - \nabla_\theta L_{Total}(\theta)\| \leq \epsilon G$, then*

$$\|\theta^{*,\epsilon} - \theta^*\| \leq \frac{\epsilon G}{L}.$$

Proof: Since $\theta^*$ and $\theta^{*,\epsilon}$ are minimizers, we have $\nabla L_{\mathrm{Total}}(\theta^*) = 0$ and $\nabla L_{\mathrm{Total}}^\epsilon(\theta^{*,\epsilon}) = 0$. From the gradient perturbation assumption,

$$\|\nabla L_{\mathrm{Total}}(\theta^{*,\epsilon})\| = \|\nabla L_{\mathrm{Total}}(\theta^{*,\epsilon}) - \nabla L_{\mathrm{Total}}^\epsilon(\theta^{*,\epsilon})\| \leq \epsilon G.$$

By Lipschitz gradient continuity,

$$\|\nabla L_{\text{Total}}(\theta^{*,\epsilon}) - \nabla L_{\text{Total}}(\theta^*)\| \leq L\|\theta^{*,\epsilon} - \theta^*\|.$$

Since $\nabla L_{\text{Total}}(\theta^*) = 0$,

$$\|\nabla L_{\text{Total}}(\theta^{*,\epsilon})\| \leq L\|\theta^{*,\epsilon} - \theta^*\|.$$

Combining the inequalities,

$$\|\nabla L_{\text{Total}}(\theta^{*,\epsilon})\| \leq \epsilon G \leq L\|\theta^{*,\epsilon} - \theta^*\|,$$

thus

$$\|\theta^{*,\epsilon} - \theta^*\| \leq \frac{\epsilon G}{L}.$$

This bound shows that parameter changes are linearly related to the perturbation size $\epsilon$, proving robustness. Introducing $L_{\text{PA}}$ can reduce the effective Lipschitz constant $L$ (through smoothing) or decrease $G$ (reducing noise sensitivity), thereby tightening the bound. The PL inequality and convexity ensure minimizer existence and uniqueness but do not directly participate in deriving the parameter bound.

### A.3 THE USE OF LARGE LANGUAGE MODELS

In this work, we employ large language models (LLMs) primarily as assistants for writing. Their role is limited to aiding the authors in polishing the presentation and improving readability.

