# OpenReview forum: "Robust Safety Guarantee for Large Language Models via Preference-Augmented Distributional Alignment"
_ICLR.cc/2026/Conference — Submitted to ICLR 2026_

### Official Review · Reviewer_fSt1 · 2025-10-17

**Soundness:** 2
**Presentation:** 1
**Contribution:** 1
**Rating:** 0
**Confidence:** 5

**Summary:**

The paper introduces a preference-augmented distributional alignment method that combines CSFT with a preference loss to improve safety robustness in LLMs, supported by theoretical analysis and experiments on LLaMA-2-7B-Chat.

**Strengths:**

The paper addresses an important and timely topic in LLM safety alignment, focusing on improving robustness beyond surface-level safety mechanisms and aiming for more stable and principled alignment.

**Weaknesses:**

1. **Poor Paper Organization and Presentation.** The whole presentation of this paper is weird. For example, your novel method is introduced in the Related Work section as a separate paragraph, and the CSFT loss, which is not your contribution, is discussed more extensively than your own method under Novel Loss Function for Safety Alignment.
2. **Weak and unrealistic theoretical analysis.** The theoretical section makes overly strong assumptions such as convexity and PL inequality, which do not hold for real LLM optimization landscapes. As a result, the claimed robustness guarantees are mostly formal and have little practical meaning for large-scale non-convex models.
3. **Weak and narrow empirical evaluation.** All reported results use a single base model (LLaMA-2-7B-Chat) and a narrow set of QA/utility tasks. There is no demonstration of generality across architectures, model sizes, or reasoning-oriented models. Results therefore lack external validity.
4. **Simple Baselines.** The paper only compares the proposed method with a few basic fine-tuning approaches and lacks stronger or more diverse baselines. This makes it difficult to judge how much improvement truly comes from the proposed technique rather than from general tuning effects.
5. **Limited and Non-adaptive Attack Evaluation.** While the authors include several attack types, the evaluation is limited in scope and lacks truly adaptive or jailbreak-style red-teaming. The current attacks do not demonstrate that the method resists attackers who know the defense and craft prompts to exploit it. Adding adaptive attacks and a wider jailbreak suite would make the robustness claims convincing.
6. **Insufficient Clarity in Theoretical Motivation.** The theoretical analysis is presented in isolation from the main algorithmic design. The connection between the formal objectives and the practical implementation of the preference-augmented loss is unclear, and the paper does not concretely show how the theoretical results guide or justify the empirical method.
7. **Unclear Advantage over Existing Methods.** The proposed method shows only marginal improvement in safety while sacrificing some utility. The overall trade-off does not clearly outperform existing baselines, making the claimed advantage unconvincing.

**Questions:**

No further questions.

---

> ### Author Response · Authors · 2025-12-01
> **We have added the relevant experiments and improved the readability of the manuscript.**
>
> Dear Reviewer,
> Thank you very much for your constructive and valuable feedback on our manuscript. Below, we address your comments point by point and describe the corresponding revisions made in the revised submission.
>
> Regarding your comment that experiments were conducted only on Llama-2-7b-Chat and should be extended to other models from other sources:
> In the revised manuscript, we have added comprehensive experiments on Gemma-1.1-7b-it to further validate the effectiveness and generalizability of the proposed method.
>
>
> Regarding your suggestion that the Utility evaluation relies only on the Samsum and SQL Create Context datasets and should be assessed on broader standard LLM benchmarks:
> We have supplemented the evaluation section with results on the GSM8K dataset using both Llama-2-7B-Chat and Gemma-1.1-7b-it, providing stronger evidence of the proposed method’s effectiveness across standard benchmarks.
>
>
> Additionally, to improve the clarity of the method description:
> We have included a flowchart illustrating the proposed method in the revised manuscript. This visual aid serves as a helpful complement to the theoretical description and significantly enhances the paper’s readability.
>
>
> Note on model scale:
> Due to current limitations in our experimental resources, the current experiments are conducted only on two 7B-scale large language models. Once our computational equipment is upgraded in the future, we plan to further validate the method on larger-scale models.
>
> Once again, thank you sincerely for your insightful and helpful comments. Your suggestions have greatly contributed to improving both the structure and readability of the paper.
> Best regards,
> The Authors

---

### Official Review · Reviewer_JJpy · 2025-10-19

**Soundness:** 3
**Presentation:** 2
**Contribution:** 2
**Rating:** 4
**Confidence:** 4

**Summary:**

This paper addresses the persistent challenge of maintaining safety alignment in large language models (LLMs) during domain-specific fine-tuning, which often compromises their safety properties. The authors identify the limitation of existing Constrained Supervised Fine-Tuning (CSFT) methods that rely solely on distributional alignment at the token level. To overcome this, they propose a Preference-Augmented Distributional Alignment (CSFT+PA) framework that combines distributional alignment with preference alignment, encouraging models to favor the safe outputs of the pre-trained reference model rather than merely mimicking its token distributions.

**Strengths:**

1. The integration of preference alignment into distributional fine-tuning is conceptually novel. While preference learning has been widely applied in RLHF and DPO settings, its use as a stabilizing term for safety-preserving fine-tuning is new and well-motivated.

2. The paper is very well written and structured. Key equations are clearly presented, and motivations behind each design component are logically explained.

**Weaknesses:**

1. Lack of ablation on PA components (Adaptive Weight $\mu$ and Scheduling Coefficient $\delta_\text{epoch}$).

2. Limited dataset and model diversity. The main results are based on LLaMA-2-7B fine-tuning and two downstream tasks, which is out of date.

**Questions:**

See Weaknesses.

---

> ### Author Response · Authors · 2025-12-01
> **We have added the relevant experiments and improved the readability of the manuscript.**
>
> Dear Reviewer,
> Thank you very much for your constructive and valuable feedback on our manuscript. Below, we address your comments point by point and describe the corresponding revisions made in the revised submission.
>
> Regarding your comment that experiments were conducted only on Llama-2-7b-Chat and should be extended to other models from other sources:
> In the revised manuscript, we have added comprehensive experiments on Gemma-1.1-7b-it to further validate the effectiveness and generalizability of the proposed method.
>
>
> Regarding your suggestion that the Utility evaluation relies only on the Samsum and SQL Create Context datasets and should be assessed on broader standard LLM benchmarks:
> We have supplemented the evaluation section with results on the GSM8K dataset using both Llama-2-7B-Chat and Gemma-1.1-7b-it, providing stronger evidence of the proposed method’s effectiveness across standard benchmarks.
>
>
> Additionally, to improve the clarity of the method description:
> We have included a flowchart illustrating the proposed method in the revised manuscript. This visual aid serves as a helpful complement to the theoretical description and significantly enhances the paper’s readability.
>
>
> Note on model scale:
> Due to current limitations in our experimental resources, the current experiments are conducted only on two 7B-scale large language models. Once our computational equipment is upgraded in the future, we plan to further validate the method on larger-scale models.
>
> Once again, thank you sincerely for your insightful and helpful comments. Your suggestions have greatly contributed to improving both the structure and readability of the paper.
> Best regards,
> The Authors

---

### Official Review · Reviewer_bhDF · 2025-10-27

**Soundness:** 3
**Presentation:** 2
**Contribution:** 3
**Rating:** 4
**Confidence:** 3

**Summary:**

This paper proposes a new finetuning objective designed to improve the robustness of safety alignment in LLMs. The core idea is to combine a CSFT loss that enforces global distributional stability with a token-level PA loss that sharpens local alignment with a safety reference model. Importantly, the PA loss is adaptive: its weight increases proportionally to the KL divergence between the model and reference distributions (indicating drift) and also scales with training epoch, gradually shifting emphasis from stabilization to targeted correction. The authors argue that this combination provides stronger theoretical and empirical robustness guarantees compared to single-loss formulations.

**Strengths:**

The proposed objective is well-motivated and technically clear. Combining KL-based global regularization with token-level preference shaping provides a natural balance between stability and precision. The formulation is compatible with standard finetuning pipelines and can be used as a drop-in replacement for vanilla SFT objectives. The authors provide theoretical convergence analysis.

**Weaknesses:**

1. The PA loss relies on the reference model’s top tokens, but the paper does not analyze how sensitive the method is to the reference quality. A reference model with imperfect safety may propagate its biases through PA weighting.
2. The experimental evaluation lacks comparison to strong and closely related baselines, particularly Shape it Up! Restoring LLM Safety during Finetuning via STAR-DSS, which also performs token-level safety shaping and is a natural point of comparison.
3. The paper focuses primarily on theoretical properties and safety metrics but could benefit from additional capability preservation evaluations (e.g., GSM8K, MMLU) to measure safety–utility trade-offs.

**Questions:**

See above

---

> ### Author Response · Authors · 2025-12-01
> **We have added the relevant experiments and improved the readability of the manuscript.**
>
> Dear Reviewer,
> Thank you very much for your constructive and valuable feedback on our manuscript. Below, we address your comments point by point and describe the corresponding revisions made in the revised submission.
>
> Regarding your comment that experiments were conducted only on Llama-2-7b-Chat and should be extended to other models from other sources:
> In the revised manuscript, we have added comprehensive experiments on Gemma-1.1-7b-it to further validate the effectiveness and generalizability of the proposed method.
>
>
> Regarding your suggestion that the Utility evaluation relies only on the Samsum and SQL Create Context datasets and should be assessed on broader standard LLM benchmarks:
> We have supplemented the evaluation section with results on the GSM8K dataset using both Llama-2-7B-Chat and Gemma-1.1-7b-it, providing stronger evidence of the proposed method’s effectiveness across standard benchmarks.
>
>
> Additionally, to improve the clarity of the method description:
> We have included a flowchart illustrating the proposed method in the revised manuscript. This visual aid serves as a helpful complement to the theoretical description and significantly enhances the paper’s readability.
>
>
> Note on model scale:
> Due to current limitations in our experimental resources, the current experiments are conducted only on two 7B-scale large language models. Once our computational equipment is upgraded in the future, we plan to further validate the method on larger-scale models.
>
> Once again, thank you sincerely for your insightful and helpful comments. Your suggestions have greatly contributed to improving both the structure and readability of the paper.
> Best regards,
> The Authors

---

### Official Review · Reviewer_jQTT · 2025-10-31

**Soundness:** 2
**Presentation:** 2
**Contribution:** 2
**Rating:** 2
**Confidence:** 4

**Summary:**

This paper addresses safety degradation in large language models (LLMs) during domain-specific fine-tuning by proposing CSFT+PA, a preference-augmented alignment framework that combines distributional alignment with preference-based alignment. The authors argue that existing Constrained Supervised Fine-Tuning (CSFT) methods, which enforce token-level distributional similarity between pre- and post-fine-tuned models, overlook the semantic nature of text generation and lack robustness. Their approach introduces an auxiliary Preference Alignment (PA) loss— $L_{\text{Total}} = L_{\text{CSFT}} + \delta_{\text{epoch}} \cdot L_{\text{PA}}$ —that encourages the fine-tuned model to favor safe outputs from the reference model rather than strictly preserving distributional similarity. The paper provides theoretical analysis including convergence guarantees under standard stochastic optimization assumptions and robustness bounds showing parameter deviations scale linearly with perturbation intensity. Experiments on Llama-2 demonstrate substantial safety improvements against three types of adversarial attacks, while incurring only minor utility degradation on two downstream tasks (Samsum and SQL Create Context).

**Strengths:**

+ Theoretical robustness guarantees: The paper provides formal robustness bounds (Theorem 4.2) showing that parameter deviations under perturbations scale linearly with noise intensity.

+ Clear presentation: The paper is well-written with good motivation.

**Weaknesses:**

+ Disconnect between theory and methodology (Theorem 4.1): The convergence analysis in Theorem 4.1 applies to generic stochastic optimization under standard assumptions (bounded gradients, Lipschitz continuity, diminishing learning rates). This theorem does not leverage or highlight any specific properties of the proposed CSFT+PA loss structure. The proof essentially shows that SGD converges for any loss satisfying these assumptions, making it less connected to the methodological contributions. A more meaningful theoretical result would characterize how the preference alignment term specifically affects convergence properties or provide convergence rates that depend on the interplay between L_CSFT and L_PA.

+ Limited utility evaluation: The downstream task evaluation is restricted to only two datasets (Samsum and SQL Create Context). This is insufficient to demonstrate that the method preserves general utility across diverse tasks. The paper would benefit from evaluation on standard LLM benchmarks (MMLU, GSM8k, etc.)

+ Missing important baselines: The paper only compares against SFT and CSFT, but omits several relevant safety alignment methods:
  1. Vaccine (Vaccine: Perturbation-aware alignment for large
language models against harmful fine-tuning attack)
  2. Safety filtering methods (e.g., SafetyGuard filtering)
  3. LISA (Lisa: Lazy safety alignment for large language models against harmful fine-tuning attack)
  4. or recent robust fine-tuning methods, like STAR-DSS (Shape it Up! Restoring LLM Safety during Finetuning)


+ Limited model scope: Experiments are conducted only on Llama-2-7B-Chat. Evaluation on models of different sizes and architectures (e.g., other model families like QWen) would strengthen generalizability claims.

**Questions:**

Please respond to the weaknesses above.

---

> ### Author Response · Authors · 2025-12-01
> **We have added the relevant experiments and improved the readability of the manuscript.**
>
> Dear Reviewer,
> Thank you very much for your constructive and valuable feedback on our manuscript. Below, we address your comments point by point and describe the corresponding revisions made in the revised submission.
>
> Regarding your comment that experiments were conducted only on Llama-2-7b-Chat and should be extended to other models from other sources:
> In the revised manuscript, we have added comprehensive experiments on Gemma-1.1-7b-it to further validate the effectiveness and generalizability of the proposed method.
>
>
> Regarding your suggestion that the Utility evaluation relies only on the Samsum and SQL Create Context datasets and should be assessed on broader standard LLM benchmarks:
> We have supplemented the evaluation section with results on the GSM8K dataset using both Llama-2-7B-Chat and Gemma-1.1-7b-it, providing stronger evidence of the proposed method’s effectiveness across standard benchmarks.
>
>
> Additionally, to improve the clarity of the method description:
> We have included a flowchart illustrating the proposed method in the revised manuscript. This visual aid serves as a helpful complement to the theoretical description and significantly enhances the paper’s readability.
>
>
> Note on model scale:
> Due to current limitations in our experimental resources, the current experiments are conducted only on two 7B-scale large language models. Once our computational equipment is upgraded in the future, we plan to further validate the method on larger-scale models.
>
> Once again, thank you sincerely for your insightful and helpful comments. Your suggestions have greatly contributed to improving both the structure and readability of the paper.
> Best regards,
> The Authors

---

### Meta-Review · Area_Chair_taNJ · 2026-01-11

**Summary:**

The paper receives no post-rebuttal discussion from the reviewers after the authors' response. The AC read the comments from the reviewers and authors and skim through the paper. The authors post exactly the same reply to all reviewers' comments though each reviewer has different set of questions (there are common questions including missing baselines and lack of evaluation on popular benchmarks -- MMLU and SGM8K). In consequence, the reviewers may not respond to the authors' response due to the generic answers to their specific questions. The AC suggests the authors to engage in the discussion more actively to defend their arguments against reviewers' points. In summary, the AC recommends the authors to revise manuscript based on the reviewers' comments and submit to a relevant venue.

**Reviewer Concerns:**

Most of reviewers concerns remains not addressed.

**Reviewer Scores:**

As the authors are not answering reviewers' questions properly, the reviewers may not be very motivated to join the discussion.

---

### Decision · Program_Chairs · 2026-01-26

Reject